# Novel Therapies for the Prevention of Fibrosis in Glaucoma Filtration Surgery

**DOI:** 10.3390/biomedicines11030657

**Published:** 2023-02-21

**Authors:** Christine G. Shao, Nishant R. Sinha, Rajiv R. Mohan, Aaron D. Webel

**Affiliations:** 1School of Medicine, University of Missouri, Columbia, MO 65212, USA; 2Harry S. Truman Memorial Veterans’ Hospital, Columbia, MO 65212, USA; 3One-Health Vision Research Program, Departments of Veterinary Medicine & Surgery and Biomedical Sciences, College of Veterinary Medicine, University of Missouri, Columbia, MO 65212, USA; 4Mason Eye Institute, School of Medicine, University of Missouri, Columbia, MO 65212, USA

**Keywords:** glaucoma, filtration surgery, conjunctival fibrosis, MMC

## Abstract

Conjunctival fibrosis remains the major impediment to the success of glaucoma filtration surgery. Anti-metabolites remain the gold standard for mitigating post-surgical fibrosis, but they are associated with high complication rates and surgical failure rates. Establishing a more targeted approach to attenuate conjunctival fibrosis may revolutionize the surgical approach to glaucoma. A new strategy is needed to prevent progressive tissue remodeling and formation of a fibrotic scar, subsequently increasing surgical success and reducing the prevalence of glaucoma-related vision loss. Advancements in our understanding of molecular signaling and biomechanical cues in the conjunctival tissue architecture are broadening the horizon for new therapies and biomaterials for the mitigation of fibrosis. This review aims to highlight the strategies and current state of promising future approaches for targeting fibrosis in glaucoma filtration surgery.

## 1. Introduction

Glaucoma is a progressive, degenerative disease of the optic nerve that can result in severe visual field loss and blindness. It is the leading cause of irreversible blindness worldwide [1,2,3,4,5,6]. The overall worldwide prevalence of primary open-angle glaucoma, the most common form of glaucoma, is 2.4% (95% CI 2.0~2.8%), and it affects an estimated 68 million people worldwide [7].

Currently, the only modifiable risk factor and proven therapy to prevent the progression of glaucoma is lowering the intraocular pressure (IOP). Reduction of IOP can be accomplished with topical eye drops, oral medications, or laser procedures that increase aqueous outflow or reduce aqueous production. The surgical treatment of glaucoma similarly decreases aqueous inflow or increases aqueous outflow. The latter can be accomplished by augmenting existing outflow pathways or creating artificial outflow pathways into the subconjunctival or sub-Tenon’s space, often referred to as traditional incisional glaucoma filtration surgery. Micro-invasive glaucoma surgical (MIGS) approaches allow for enhancement of the conventional aqueous outflow pathways through the trabecular meshwork and into Schlemm’s canal or alternatively from the anterior chamber into the suprachoroidal space. Traditional incisional glaucoma filtering surgery allows the aqueous humor to drain into the space between the sclera and conjunctiva and/or Tenon’s tissues.

Unlike many surgeries, where the goal is to heal tissue with the restoration of normal architecture, traditional incisional glaucoma filtering surgery aims to modulate wound healing. Wound healing modulation allows for continued aqueous egress into the subconjunctival and/or sub-Tenon’s space, thereby preventing surgical failure [8,9]. Episcleral and subconjunctival fibrosis remain major impediments to the success of glaucoma filtering surgery. Conjunctival fibrosis at the site of filtration may impede aqueous outflow, leading to inadequate IOP reduction. Surgical failure is associated with poor postoperative IOP control, consequent progression of glaucomatous disc cupping, and visual field loss [10]. 5-fluorouracil (5-FU) and mitomycin-C (MMC) are widely used to attenuate ocular fibrosis. The usage of 5-FU and MMC have led to improved surgical success rates, but with a concomitant increase in postoperative complications. Complications associated with broad non-specific antifibrotics include wound leaks, hypotony, and infection [11]. Therefore, broadening the therapeutic approach for glaucoma filtration surgery is necessary, and agents with more specific physiological actions and less cytotoxicity are needed. This review aims to highlight the current understanding of potential therapeutic targets for the prevention of fibrosis and discuss novel methods to improve glaucoma surgical success.

## 2. Causes of Ocular Fibrosis following Glaucoma Filtration Surgery

### 2.1. Wound Healing

Like other surgeries, glaucoma filtration surgery induces tissue trauma. Crosstalk among various players such as growth factors, cytokines, signaling pathways, and immune regulators is critical during conjunctival wound healing. Wound healing has been defined in three distinct and overlapping phases: the inflammatory, proliferative, and maturation stages (Figure 1) [10,12,13,14]. In the inflammatory phase, plasma proteins and blood cells migrate to the site of injury. During this phase, damaged cells and pathogens are removed while bleeding is controlled. The resultant clot, platelet plug hormones, cytokines, and growth factors attract neutrophils, macrophages, and lymphocytes to the site of injury. IL-1, Interferon-α2b, and growth factors such as transforming growth factor β (TGFβ), vascular endothelial growth factor (VEGF), and platelet-derived growth factor (PDGF) are among these mediators [10].

In the proliferative phase, endothelial cells and fibroblasts migrate to the site of injury. The key processes in this phase are angiogenesis and granulation tissue formation. Fibroblasts increase in number, while neutrophils and monocytes secrete proteolytic enzymes and promote debridement. Activated phagocytes increase growth factors and cytokines such as TGFβ to activate and maintain fibroblasts. VEGF promotes blood vessel formation, and PDGF stimulates fibroblasts [12,13]. 

In the maturation phase, scar tissue is formed during tissue remodeling as blood vessels regress over time and fibroblasts induce crosslinking of collagen type I and elastin. This phase is defined by matrix metalloproteinases (MMPs), which are synthesized by macrophages, neutrophils, and fibroblasts. This synthesis leads to collagen supercoils and the transformation of granulation tissue into scar tissue [10]. The reduction in myofibroblast number is also key to this phase, as their prolonged survival leads to excessive scarring [13]. Modulation of scarring may target many points of fibrosis regulation, including the migration, proliferation, myofibroblast transdifferentiation, and apoptosis of fibroblasts [15].

### 2.2. Growth Factors and Cytokines

TGFβ, PDGF, and VEGF are cytokines that play a key role in modulating ocular fibrosis [16]. Tissue damage transforms TGFβ into its active form. Fibroblasts migrate to the surgical site and are transformed into extracellular matrix (ECM)-producing myofibroblasts, the precursors for Tenon’s fibroblasts, the most important mediator of the development of conjunctival fibrosis under the scleral flap. TGFβ is a potent inducer of Tenon’s fibroblast proliferation, migration, and collagen production. TGFβ also stimulates angiogenesis and acts as a chemoattractant to other macrophages and fibroblasts [17]. PDGF induces the proliferation of fibroblasts and migration to the site of injury [18], and it is upregulated in fibrosis. VEGF plays a key role in promoting cell proliferation. It is a potent mediator of angiogenesis, vasculogenesis, and vascular endothelial cell permeability, all of which are critical processes in wound healing. The signaling pathways are discussed in the next section. 

### 2.3. Intracellular Signaling Driving Fibrosis

#### 2.3.1. TGFβ

Signal transduction through the effects of TGFβ has been extensively studied. TGFβ has been targeted for its inflammatory and proliferative effects. TGFβ is found in high concentrations in the aqueous humor of glaucoma patients [19,20] and in the trabecular meshwork [21]. Thus, it is likely that it has a role in the regulation of aqueous outflow. Several studies have shown that TGFβ activates fibroblast proliferation in vitro and in vivo [14,22,23,24]. Roberts et al. demonstrated the proliferative effects of TGFβ by injecting TGFβ into murine models, which resulted in fibroblast proliferation and collagen deposition [23].

TGFβ2 is the predominant isoform found in the eye and accounts for the greatest stimulation of fibrosis. Due to the robust amount of TGFβ in fibroblasts, targeting this pathway holds promise for improving outcomes of glaucoma surgery. However, blocking all TGFβ isoforms is excessive, as some TGFβ is essential for normal wound healing. Alternatively, the use of a monoclonal antibody directed solely against TGFβ2 may be too narrow of an approach due to the involvement of other TGFβ isoforms in ocular fibrosis.

TGFβ exerts its effects through serine/threonine and tyrosine kinase receptors. TGFβ signals through small mothers against decapentaplegic (SMAD)-independent and SMAD-dependent pathways [25]. Crosstalk between these pathways has been reported. In SMAD-dependent signaling, SMAD proteins transduce signals from the membrane to the nucleus. SMAD proteins are known to mediate the transcription of various genes for the transdifferentiation of fibroblasts to myofibroblasts [26].

SMAD 2, 3, and 4 proteins are specific for TGFβ signaling [27,28]. SMAD 2 and 3 are phosphorylated by the TGFβ type I receptor kinase and partner with SMAD 4. This complex is translocated to the nucleus, where transcription products are targeted. SMAD 3 is crucial to wound healing in many tissues. SMAD 7 is inhibitory. It is induced by TGFβ as well as other cytokines, and it blocks SMAD 2 and SMAD 3 signaling [29].

In SMAD-independent signaling, JNK, p38, mTOR, and ROCK pathways are responsible for apoptosis and actin cytoskeleton reorganization. P38 MAPK is also critical for cell migration. JNK also mediates connective tissue growth factor (CTGF) expression [30]. 

#### 2.3.2. PDGF

PDGF is another cytokine implicated in the progression of wound healing. PDGF is a dimer that can exist as a homo- or heterodimer made up of A and B subunits. PDGF stimulates keratocyte proliferation, survival, and migration. Thus, PDGF upregulation has been associated with pathological angiogenesis and fibrotic conditions. PDGF acts synergistically with TGFβ in an autocrine manner to stimulate differentiation of the myofibroblast [31]. 

#### 2.3.3. FGF2

FGF2 also promotes the proliferation and differentiation of keratocytes into fibroblasts [31] and is a possible therapeutic target to inhibit fibrosis. Unlike PDGF, FGF2 downregulates alpha-smooth muscle actin (α-SMA), which allows for wound contraction, in myofibroblasts [32].

### 2.4. Biomechanical Cues

In addition to biochemical modulation, biomechanical cues play a significant role in scar formation. Increased tissue stiffness facilitates myofibroblast transdifferentiation and fibrosis [14]. Morphologically, myofibroblasts have a contractile apparatus that contains bundles of actin microfilaments with contractile proteins. A distinguishing feature of the myofibroblast is the expression of α-SMA. The α-SMA adds to the mechanical strength of the cell and allows for wound contraction. This provides a mechano-transduction system, as the force that is generated by the fibers can be transmitted to the ECM. In addition, extracellular mechanical signals can be transduced into intracellular signals through this mechano-transduction system. Topographic features and substrate compliance are all involved in the modulation of myofibroblast differentiation. Increased expression of α-SMA is correlated with increased force generation by myofibroblasts. It has subsequently been shown that most myofibroblasts express α-SMA, and the expression of α-SMA and collagen type I in these cells is coordinated by TGFβ1. Thus, the myofibroblast plays a key role in the synthesis of ECM and force generation, which results in ECM reorganization and wound contraction [33].

It is widely accepted that the cellular response is governed by the distribution of mechanical forces throughout the tissue. Although the mechanosensitive cellular components are important to the mechanical responses in fibrosis, understanding the context of the response to glaucoma surgery may help to target new therapies in addressing the high failure and complication rates in this surgery [34].

Coordinating the response from physical force to biochemical information is essential to tissue healing [35]. Force can change the response to glaucoma filtration surgery in that wall stress is proportional to capsule thickness in glaucoma surgery [36]. Increased tissue stiffness facilitates myofibroblast transdifferentiation and subsequent fibrosis. Similarly, after phototherapeutic keratectomy, elevated stiffness of the anterior corneal stroma precedes the upregulation of myofibroblasts, and this stiffness does not return to pre-surgical values [37]. In vitro, stromal cells treated with TGFβ1 were stiffer than untreated cells. The stiffness of these cells has been shown to depend on the substrate stiffness. Thus, the geometry of biomaterials and implantable devices impacts the extent of fibrosis, and the stiffness of the microenvironment surrounding the incision can change cellular mechanics [36,37]. The remodeling of ECM fibers by fibroblasts in interstitial flow also contributes to the extent of fibrosis, as a perpendicular alignment can also decrease shear and drag forces, decrease tissue permeability, and increase hydrostatic pressure [38]. The filtering bleb after filtration surgery is subjected to mechanical stress from the draining of aqueous humor. Further investigation into the integration between biomechanical and biochemical signaling is warranted. These studies will allow for the development of new therapies that consider the biochemical and biophysical microenvironment to mediate ocular fibrosis.

Ultimately, further understanding of the molecular patterns and biomechanics involved in conjunctival wound healing will allow for more targeted therapies to reduce IOP and prevent glaucoma-related complications.

## 3. Strategies to Prevent Ocular Fibrosis following Glaucoma Filtration Surgery

Although limiting intraoperative bleeding may attenuate fibroblast activity, this is insufficient to reduce long-term fibrosis. The following section discusses current and promising new therapies in development, which are summarized in Table 1.

### 3.1. Modulating Inflammation

Numerous anti-inflammatory agents including steroids and NSAIDs are used clinically to increase surgical success and modulate wound healing. Intraoperative and postoperative steroids can increase the success of glaucoma surgery [10,13]. Steroid use is mediated by the suppression of leukocyte concentration, distribution, and function. It ultimately diminishes the fibroblast activity in addition to the vascular effects. The reduction of neutrophils at the inflammatory site shifts lymphocytes, monocytes, and basophils out of circulation and into lymphoid tissue. Consequently, steroid inhibition of leukocyte concentration and function as well as vascular permeability led to less local tissue disruption, reduction in the amounts and activities of mitogens and growth factors, and decreased fibrin production, all of which result in attenuation of wound healing [13]. Topical steroids are often administered, but different dosages, routes, and durations of therapy have resulted in variable outcomes [104]. Topical corticosteroids use following trabeculectomy appears effective, but there has been no definitive consensus on dosing and duration of treatment [104]. The main challenges with steroid use are the systemic side effects, short aqueous half-life, prolonging its release, and the possibility of cataractogenesis with prolonged usage [105,106].

Although the usage of topical NSAIDs may enhance the lowering of IOP from topical prostaglandin analog treatment for glaucoma patients, the usage of NSAIDs with glaucoma filtration surgery remains controversial [107]. Clinically, NSAIDs are not routinely used after glaucoma filtration surgery. NSAIDs exhibit non-inferiority to steroids in terms of postoperative IOP control. A 2019 meta-analysis showed insufficient evidence to recommend one over the other [108]. Although most studies in the meta-analysis suggest no difference in the IOP-lowering ability of NSAIDs and steroids, other studies show conflicting evidence. A study by Yuen et al. compared topical dexamethasone to ketorolac after filtering glaucoma surgery with Ahmed glaucoma valve implantation. They found that mean IOP was significantly greater in the steroid group only at week 4 postoperatively, while the difference was not statistically significant at all other time points. In addition, NSAIDs showed a greater incidence of conjunctival retraction compared to dexamethasone [109]. However, a randomized controlled trial (RCT) by Breusegem et al. showed that topical ketorolac for 1 month preoperatively improved trabeculectomy outcomes when measured by the likelihood of postoperative needling. However, when ketorolac was compared to the steroid fluorometholone administered preoperatively, ketorolac was not as effective in reducing the need for additional postoperative IOP-lowering medications to reach the target IOP [57]. A retrospective cohort study showed that patients with filled NSAID prescriptions in the perioperative period had fewer interventions for bleb failure compared to patients without filled NSAID prescriptions [58]. These data suggest that steroids and NSAIDs may have differing effects on wound modulation following surgery, but our current understanding is that one is not necessarily superior to the other. Due to the limited and contradicting evidence, more investigation on this topic is necessary. 

### 3.2. Modulating Would Healing

#### 3.2.1. 5-FU

5-FU is a chemotherapeutic agent, and its antiproliferative effect stems from its ability to antagonize pyrimidine metabolism. Hence, DNA synthesis is inhibited, and cells die in the S-phase. 5-FU is effective at inhibiting human and animal fibroblasts in addition to inhibiting fibroblast-mediated collagen contraction. However, 5-FU is toxic to actively replicating tissues [39,79,110]. There is potential for the reduction of filtering surgical failure when injected through the conjunctiva postoperatively [40]; however, the increased number of visits, the discomfort of repeated injections, and epithelial toxicity are often cited as reasons why 5-FU is most commonly used on an ad hoc basis or in cases of imminent bleb failure [104]. In a study by Wormald et al., no evidence was found of an increased risk of serious sight-threatening complications, but common complications included hypotony maculopathy and wound leak [41].

#### 3.2.2. MMC

MMC is a DNA-crosslinking-alkylating agent that inhibits the cell cycle, DNA replication, cell mitosis, and protein synthesis. MMC intercalates DNA and targets cells throughout the cell cycle to block cell proliferation. Many clinical studies have shown that MMC reduces scar formation, increases the rate of bleb survival, and maintains IOP reduction [42]. 

MMC is routinely used with subconjunctival/sub-Tenon’s glaucoma filtration surgery, such as trabeculectomy and deep sclerectomy. It is also used with newer subconjunctival MIGS procedures, including the XEN gel stent or Preserflo microshunt surgery, to prevent fibrosis and increase surgical success rates. Complications are often associated with antimetabolite treatment, including hypotony, blebitis, endophthalmitis, bleb leakage, and vision loss [111,112,113,114,115]. More detail on the innovations and history of MMC use has been provided previously [116]. 

Although the method of application may affect the rate of surgical failure and complications, multiple studies have found that it does not result in significant differences. In a study looking at Asian eyes, phacoemulsification-trabeculectomy combined with subconjunctival MMC injection had comparable outcomes to that of sponge-applied MMC, with a similar reduction in IOP at 1, 6, and 12 months postoperatively and a lower postoperative complication rate [43]. However, MMC injection may lower the need for clinic visits within the first 3 postoperative months and reduce the need for 5-FU intervention when compared to MMC sponge application [117]. In an RCT, Do et al. demonstrated that preoperative subconjunctival injection of MMC and intraoperative direct scleral application of MMC with surgical sponges resulted in comparable surgical outcomes [118]. However, some studies suggest that sub-Tenon injection of MMC produces more favorable bleb morphology after trabeculectomy [44,45,46]. Methods and dosage articles have been recently reviewed elsewhere by Bell et al. Due to the greater potency of MMC, intraoperative application of MMC has become popular with adjunctive postoperative 5-FU [42].

In an RCT, there were no differences in the types of complications from 5-FU and MMC, as both can contribute to bleb leakage, hypotony, choroidal effusion, and endophthalmitis in the short to medium term [119]. Though MMC inhibits fibroblast proliferation more effectively and permanently than 5-FU, it may be associated with higher rates of complications [80,81,82]. Experimental evidence and clinical use indicate that intraoperative MMC can cause pale, avascular blebs that can become thin and lead to epithelial surface breakdown. This progression is due to the MMC-induced apoptosis of conjunctival connective fibroblasts [120]. The toxic effects of MMC may be mitigated by combining a lower dosage of MMC with valproic acid, which has been demonstrated in a rabbit model [121,122]. There is no definite clinical recommendation for post-operative 5-FU as opposed to MMC.

#### 3.2.3. Beta Radiation

In vitro studies have shown that beta radiation diminishes fibroblast proliferation, likely through a mechanism involving increased p53, a key enzyme in the cell cycle [83,84]. Kirwan et al. compared trabeculectomy with beta radiation to trabeculectomy without beta radiation across studies that included black African, Caucasian, and Chinese subjects and concluded the same findings in RCTs [47]. At the time of trabeculectomy, beta radiation can minimize scar tissue formation and increase the likelihood that surgery will effectively lower the IOP, but it has been shown to increase the risk of cataract formation [47]. In vitro studies have shown that beta radiation targets ECM production as opposed to fibroblast migration or contraction [84]. The advantages of beta radiation include ease of application, cost, and accessibility. Disadvantages of beta radiation include the possibility of cataract formation and keratopathy. In an RCT looking at trabeculectomies, beta radiation with trabeculectomy led to less bleb failure and reduced IOP compared with MMC with trabeculectomy in the South African population [123]. Another study in Egyptian eyes showed that preoperative beta radiation in addition to intraoperative MMC patients during trabeculectomy had greater IOP control. However, beta radiation may increase the risk of a cystic bleb. Beta radiation shows promise in increasing glaucoma filtration surgery success; however, there is not enough evidence to recommend beta radiation over current clinical standards of practice. Large, multicenter RCTs comparing beta radiation and MMC in glaucoma filtration surgery are warranted [48]. 

Overall, there is great variation in the usage of anti-scarring adjuncts before, during, and/or after surgery, and many adjuncts can be associated with a higher complication rate. A greater understanding of the cellular mechanisms of the wound healing response has led to the identification and modulation of potential new therapeutic targets.

#### 3.2.4. Anti-TGFβ Agents

Monoclonal antibodies directed at TGFβ are advantageous due to their specificity. One of the key advantages of an antibody is its target specificity as opposed to a nonspecific mechanism of action, as in the case of 5-FU and MMC. Lerdelimumab is a monoclonal antibody to TGFβ2. Even though it was promising following laboratory studies, a human RCT failed to show the efficacy of Lerdelimumab in preventing the failure of trabeculectomy [124]. Fresolimnumab is a monoclonal antibody that neutralizes all forms of TGFβ; however, to our knowledge, it has not been tested in the setting of ocular fibrosis [85,86]. Further investigation into the use of monoclonal antibodies to TGFβ in glaucoma filtering surgery is warranted.

Targeting SMAD proteins by overexpressing inhibitory SMAD7 or reducing SMAD2, 3, or 4 can reduce stromal fibrosis [26]. In vivo studies have shown that expression of exogenous SMAD7 in burned corneal tissue results in reduced activation of SMAD signaling [125]. In work by Yamanaka et al., gene transfer of SMAD7 suppressed fibrogenic and inflammatory signals, including α-SMA, VEGF, and CTGF [59]. In work by Gupta et al., topical application of SMAD7 through recombinant adeno-associated virus serotype 5 to rabbit cornea post-photorefractive keratectomy resulted in a decrease in corneal fibrosis [126]. Topographical cues in the biophysical environment have also been shown to increase the expression of SMAD7 to prevent fibrosis by modulating TGFβ-induced myofibroblast differentiation and α-SMA expression [126]. To our knowledge, directly targeting SMAD proteins has not been studied in glaucoma filtration surgery, but this strategy may have therapeutic potential in the prevention of excess scarring. 

Fibroblasts in the cornea express TGFβ induced factors (TGIF) 1 and 2. TGIFs repress SMAD-dependent signaling at the level of transcription. In vitro studies have demonstrated that TGIF1 gene editing via CRISPR/Cas9 can attenuate myofibroblast formation and protein levels of profibrotic genes [127]. Vorinostat is a histone deacetylase inhibitor that increases TGIF levels to inhibit TGFβ profibrotic expression of genes [91,92]. Vorinostat has been studied in animal models of trabeculectomy and is discussed in Section 3.2.12. Trichostatin and ITF2357 are other histone deacetylase inhibitors that can also affect SMAD signaling [93,94]. Trichostatin A has been shown to decrease bleb vascularity, leukocyte infiltration, and expression of α-SMA and TGF-β1 in the conjunctiva in rat models of trabeculectomy [128]. ITF2357 has been shown to attenuate corneal fibrosis in vivo but has not been studied in models of glaucoma filtration surgery.

SMAD-independent pathways may also be targeted through PPARγ ligands, p38 MAPK inhibitors, JNK inhibitors, mTOR (PI3K) inhibitors, and Rho signaling inhibitors. The PPARγ ligand Rosiglitazone effectively controls corneal fibrosis in vivo and in vitro, while pioglitazone has been shown to modulate fibrosis in vitro [30,129,130,131]. Studies have shown promise for p38 MAPK inhibitors in vitro and JNK inhibitors in a rat model [30,124,132]. The mTOR inhibitor Rapamycin has been shown to reduce corneal scarring in a rabbit model [133,134]. The Rho family of small GTPases orchestrate fundamental biological processes, including cell migration and actin cytoskeleton dynamics [135,136]. ROCKs are Rho effectors that play an important role in regulating stress fiber formation in fibroblasts and epithelial cells in vitro [137]. ROCK inhibitors have been found to attenuate fibroproliferation and scar formation in a rabbit model of glaucoma surgery, which may be due to decreased matrix contraction, thereby reducing scarring [136]. 

While TGFβ2 is the predominant isoform of TGFβ in the eye and the dominant isoform in conjunctival fibrosis, TGFβ1 is the dominant mediator of the corneal fibrotic response [138]. TGFβ1 is involved in ECM protein expression and pathological tissue fibrosis [139]. In vitro studies have shown that KCa3.1 (calcium-activated potassium channel) inhibitor TRAM34 reduced TGFβ-mediated cell migration and myofibroblast formation in human corneal fibroblasts [140]. CTGF, a regulatory molecule in the TGFβ pathway, is another target for mediating scarring, as it is involved with ECM production, actin cytoskeletal dynamics, and contractile force in human subconjunctival fibroblasts. A study by Lee et al. showed that CTGF suppression by CRISPR in a rabbit glaucoma filtration surgery model showed significantly better survival at the surgery site, less subconjunctival fibrosis, limited collagen deposition, and reduced cellularity when compared with untreated eyes [141]. 

Pirfenidone is another novel agent that has shown antifibrotic and antiangiogenic potential, which has been established by many in vitro and in vivo studies [63,95,96]. It is currently implicated in treating idiopathic pulmonary fibrosis [97]. It is possible that pirfenidone inhibits TGFβ and reduces TNFα, but the mechanism of action is unknown [142]. In vitro, pirfenidone inhibits extracellular matrix deposition in ocular fibroblasts and decreases fibroblast proliferation, migration, and contraction [143,144]. It has been shown that 0.5% pirfenidone eye drops improve trabeculectomy bleb success in a rabbit model [64].

#### 3.2.5. Anti-YAP/TAZ

In vitro studies have found the Yes-associated protein (YAP)/transcriptional co-activator PDZ-binding motif (TAZ) to be involved in the activation of SMAD2/3. YAP/TAZ is a potential target in TGFβ2–mediated conjunctival fibrosis and TGFβ1-induced myofibroblast transformation [87,88]. YAP/TAZ interacts with SMAD proteins and likely transduces stiffness to the myofibroblast phenotype in the cornea [138]. Verteporfin is a YAP/TAZ inhibitor, and in vitro studies have found that it may improve outcomes in glaucoma surgery by suppressing TGFβ2-YAP/TAZ-SMAD signaling [87]. Therefore, YAP/TAZ may act as orchestrating molecules that integrate the biophysical and biochemical responses. 

Other evolving approaches to targeting the TGFβ pathway include long non-coding RNAs, which is a strategy that silences gene expression. MicroRNAs (miRNAs) are non-coding RNAs that regulate gene expression at the post-transcriptional level by base-pairing with complementary sequences of the mRNA, which cleaves or destabilizes the mRNA. A study by Hwang et al. found that miRNA 143/1145 inhibition in human Tenon’s fibroblasts may be a promising target for subconjunctival fibrosis [60]. Overexpression of miRNA 26a and 26b may also be targeted, as they significantly inhibit lens fibrosis in vitro and in vivo [145].

#### 3.2.6. Anti-SPARC

Secreted protein acidic and rich in cysteine (SPARC) is a matricellular protein involved in ECM production and organization that has been shown to be upregulated by TGFβ2 through a post-transcriptional mechanism [89,90]. Seet et al. showed that silencing SPARC in Tenon’s fibroblasts reduces profibrotic gene expression without stimulating apoptosis, as with MMC [146]. 

SPARC knock-out mice have shown improved surgical survival due to less collagenous ECM and smaller collagen fibril diameter [61]. To target this gene in humans, gene therapies involving siRNAs are in development. Nanoparticles serve as a vehicle to transport ligands such as antibodies and peptides into the cell for gene therapy. Nanoparticle-based siRNA delivery has been tested to target the SPARC gene and is still being tested in animal studies; pilot studies have shown that this method is safe and feasible [62]. Another delivery system of SPARC siRNA utilizing a hydrogel is currently in animal studies. Its safety profile is promising and shows effective inhibition of subconjunctival scarring in a rabbit model [147].

#### 3.2.7. Anti-PDGF Subunit B Agents

Studies suggest that E10030, an anti-PDGF aptamer, inhibits angiogenesis and tumor growth, but its effect in the context of ocular fibrosis has not been well-studied [65,66]. A study on aptamers ARC126 and ARC127, aptamers that bind PDGF subunit B, showed that there is potential for aptamers in reducing scarring and fibroblast proliferation in mouse models with proliferative retinopathies [148]. 

Kinase inhibitors such as Nintedanib inhibit the PDGF receptor, VEGF receptor, and FGF receptor tyrosine kinases. They reduce the TGFβ1-induced phosphorylation of SMAD2/3, p38 MAPK, and ERK1/2 to prevent Tenon’s fibroblast activation in vitro [97,149]. Thus, there are potent antifibrotic effects on Tenon’s fibroblasts, and these are potential therapeutic targets.

#### 3.2.8. Anti-VEGF Agents

Angiogenesis plays an important role in wound healing, and VEGF plays a key role in the formation of pathological blood vessels. One study showed that VEGF is expressed in aqueous humor from glaucoma patients and rabbits that have undergone surgery [67]. VEGF was also found to stimulate the proliferation of Tenon’s fibroblasts in cell cultures. In vitro, bevacizumab has been shown to inhibit fibroblast proliferation and improve scarring in an animal model by reducing angiogenesis and collagen deposition [67]. Clinical studies on IOP after using subconjunctival bevacizumab injection in glaucoma patients show equal effectiveness in IOP reduction compared with MMC and potentially a better safety profile, but bevacizumab delivered by sponge had no advantage over MMC [150,151]. Intracameral bevacizumab and MMC in trabeculectomy were shown by another group to have differences in surgical success [152]. Additionally, the anti-scarring effect of combined bevacizumab and MMC was not significantly different from MMC alone, but bevacizumab and 5-FU led to less scarring compared to either agent alone both in vitro and in vivo [153,154]. Intravitreal ranibizumab was also shown to lead to less vascularity with more diffuse blebs when compared to MMC [155]. However, there is still controversy regarding the use of anti-VEGF agents, as some studies show that there is insufficient evidence for ranibizumab and bevacizumab injection use in glaucoma surgery [145,156].

#### 3.2.9. Fibrinolytic Therapy

Tissue plasminogen activator (tPA) has been deemed useful in dissolving fibrin in blood clots after glaucoma surgery. However, hyphema, profound hypotony, and anterior chamber flattening were found in 11% of patients, though no long-term complications were associated with intracameral tPA use after a mean follow-up of 2.5 years [49]. Moreover, an RCT by Barequet et al. showed that sponge application of MMC during trabeculectomy with intracameral tPA may increase trabeculectomy success rates without a difference in safety compared to sponge application of MMC alone [157]. Urokinase is a thrombolytic agent that has been shown by one study to reduce IOP after glaucoma surgery without adverse effects when injected into the anterior chamber. However, the long-term safety of intracameral injection is uncertain [158].

#### 3.2.10. Collagen-Platelet Interaction Inhibitors

Saratin is a 12kD protein that was isolated from the saliva of the leech Hirudo medicinalis. It interferes with platelet integrin α2β1-collagen and von Willebrand factor-collagen binding, preventing platelet aggregation in response to injury [159]. Because of the interference of platelet aggregation, saratin limits PDGF, TGFβ, IGF, and EGF [99]. A single intraoperative topical application of saratin did not outperform MMC but a combination of intraoperative topical application with two additional postoperative injections did not cause bleb avascularity and tissue thinning, which is often associated with MMC treatment [68]. Although Saratin is not used clinically, further exploration into its dosing is warranted.

#### 3.2.11. LOX and LOXL Antibodies

The lysyl oxidases (LOX) are a family of enzymes that plays a role in the crosslinking of collagen and elastin [160]. Lysyl oxidase-like 2 (LOXL2) is reported to be involved in many fibrotic diseases, including lung and myocardial fibrosis [161,162]. There is also a correlation between increased LOXL2 and TGFβ levels [163]. A murine model by Adachi et al. found that LOX genes are overexpressed in the bleb region in glaucoma filtration surgery [164]. In a rabbit model of trabeculectomy, targeting LOXL2 with an inhibitory monoclonal antibody reduced pathologic angiogenesis, inflammation, and fibrosis [69]. 

#### 3.2.12. MMP Inhibitors

MMPs are proteolytic enzymes that are essential in all phases of wound healing [70]. They have an important role in wound remodeling–degrading extracellular matrix components while also being able to synthesize collagen and the extracellular matrix. MMPs are present in the bleb walls of Molteno implants of human eyes [165]. In vitro, MMP inhibition has been shown to reduce collagen contraction, cell migration, and collagen production without cellular toxicity [100]. Rabbits undergoing glaucoma filtration surgery treated with ilomastat, an MMP inhibitor, resulted in increased bleb survival, significantly lowered IOP over 30 days, and less scar tissue histologically compared to rabbits treated with the vehicle phosphate-buffered saline [70]. Topical doxycycline, another MMP inhibitor, was found to be similarly effective compared to intraoperative MMC in an in vivo rabbit glaucoma filtering surgery bleb survival study [166].

More recent preclinical tests in an animal model showed reduced inflammation and extracellular matrix remodeling with subconjunctival iIlomastat injection [71]. Preliminary results from a rabbit model show that adequate conjunctival tissue penetration and therapeutic concentrations within the sclera and conjunctiva and aqueous humor can be achieved [167].

#### 3.2.13. MRTF/SRF Inhibitors

There is increasing evidence that the serum response factor (SRF) and myocardin-related transcription factor (MRTF) play an essential role in the activation of fibroblasts [168]. The MRTF-A/SRF transcription pathway is an important upstream regulator of MMP expression in ocular fibrosis [72,169]. Because of this, the MRTF/SRF pathway represents a potential novel target to inhibit MMP expression [72,168,169].

Nanoparticle-based siRNAs have also been in development as a method to induce MRTF silencing. In vitro studies of nanoparticle-based siRNAs showed silenced expression of the MRTF-B gene in human Tenon’s fibroblasts and blocked collagen matrix contraction while not being cytotoxic [101]. In vivo studies of nanoparticle-based siRNAs doubled bleb survival and decreased conjunctival scarring, with no adverse side effects, in a rabbit model of glaucoma filtration surgery [72].

#### 3.2.14. Epigenetic Modifiers

Epigenetic modifiers regulate gene expression through methylation and acetylation of DNA and histone proteins. By altering DNA structure, histone modifiers regulate how DNA binds to its transcription factors [170]. Suberoylanilide hydroxamic acid (SAHA/vorinostat) is an HDAC inhibitor currently approved by the FDA for the treatment of cutaneous T-cell lymphoma. SAHA has been shown to improve outcomes with a pre-op subconjunctival application in a rabbit model [73]. On the other hand, a single intraoperative injection of SAHA in a rabbit model did not prolong bleb survival [171]. HDAC inhibition is another potential novel agent that can modulate glaucoma filtration surgery wound healing without the adverse effects associated with 5-FU and MMC; more investigation into its mechanism and clinical studies are necessary.

#### 3.2.15. Cell Cycle Targets

Methods of gene therapy involve increasing expressions of inhibitory genes and proteins in the cell cycle. Human p53 is a tumor suppressor gene that plays a key role in arresting cell cycle progression to allow DNA repair or inducing apoptosis if the damage is too extensive [172]. Many studies have demonstrated p53 as an active player in human Tenon’s fibroblast migration and growth [102,103]. Using a recombinant adenovirus for p53, Johnson et al. induced overexpression of p53 in human Tenon’s fibroblasts and significantly inhibited fibroblast proliferation and DNA synthesis [173].

Perkins et al. placed one of p53′s downstream effectors, p21, in a similar recombinant adenovirus vector for use in rabbit trabeculectomy. They found that eyes undergoing p21 gene therapy performed similarly to MMC in terms of decreasing IOP and preventing the proliferation of fibrosis [74]. One consideration with the use of viral vectors is the rare but inherent risk of mutagenesis. At least two cases of lymphoproliferative disorders related to gene therapy have been described in the literature.

There is also potential for therapies that target other components of the cellular division pathway, including CDKN1B. In a rabbit model of glaucoma filtration surgery, Yang et al. found that overexpression of CDKN1B, a CDK inhibitor, may reduce the severity of scar formation and improve surgical outcomes by reducing fibroblast proliferation [75].

#### 3.2.16. Broad-Spectrum Immunosuppressives

Cyclosporine is an immunosuppressive agent that prevents T-cell activation and subsequent T-cell mediated inflammation by reducing IL-2 and IL-2 receptor expression, though its benefit to glaucoma surgery outcomes is still uncertain as there is conflicting evidence. An in vivo study by Park et al. showed that postoperative topical 2% cyclosporine can enhance the effectiveness of glaucoma drainage implant surgery measured by IOP and improve the flow resistance through the implant capsule, although there was no difference in fibroblast density [76]. Recent in vivo studies with different delivery methods and concentrations have supported this data [174,175]. However, 2% cyclosporine intraoperatively and postoperatively was not associated with IOP or prolonged bleb survival in rabbits when compared with MMC [176]. In human clinical trials, topical 0.05% cyclosporine was shown to have no effect on postoperative bleb function or IOP following trabeculectomy, although there was a decrease in ocular surface disease [177]. Sirolimus is a macrolide that is used as an immunomodulatory medication that decreases the response of B and T cells and reduces the response of cytokines such as IL-2. Sirolimus applied via a poly(lactic-co-glycolic acid) sustained delivery film in a rabbit model was efficacious at preventing filtration bleb scarring and increasing the success rate of filtration surgery [178].

### 3.3. Modulating Mechanotransduction

Over the last two decades, adjuncts to conventional glaucoma surgeries have been studied to improve success rates. Because mechanotransduction initiates biochemical responses, modulating mechanical stress is imperative to maintaining the subconjunctival/sub-Tenon’s space after surgery. Physical spacers separating the subconjunctival and episcleral tissues may be used to prevent fibrosis and are still under study, as early contact between the two surfaces may lead to fibrosis and result in flat and nonfunctional blebs.

#### 3.3.1. Human Amniotic Membrane

The human amniotic membrane may be beneficial due to its anti-inflammatory, antifibrotic, and antiangiogenic properties [179]. It is integrated into the host as a surgical graft or temporarily used as a biological bandage. Two groups have shown that in vivo amniotic membrane transplantation (AMT) in the construction of filtering blebs attenuates the healing response. However, the improvement in bleb survival must be weighed against the association with delayed healing [77]. Sheha et al. compared trabeculectomy with MMC to trabeculectomy with MMC and AMT. They found that the latter group had higher success rates, lower postoperative IOPs, and fewer complications. However, a 2021 study by Roque et al. found that trabeculectomy combined with MMC and AMT did not show better results than trabeculectomy with MMC alone [180]. A 2015 Cochrane Review found 18 studies investigating the use of AMT with trabeculectomy. The review noted the data as low-quality but concluded that AMT can improve IOP slightly and may lead to fewer complications [181]. A systematic review of five RCTs showed that mean IOP was lower and that complications, including a flat anterior chamber and hyphema, were decreased in trabeculectomy with AMT compared to trabeculectomy without AMT [182].

#### 3.3.2. Perfluoropropane Gas

Perfluoropropane (C3F8) expansile gas has promise in acting as a spacer in the augmentation of glaucoma-filtering blebs. It can be injected into the subconjunctival space at the end of surgery and lasts for 2–3 weeks. Animal studies and pilot human studies have shown safe and promising results in the early post-op period as well as long-term IOP control [50]. Thus, injection of C3F8 gas at the time of glaucoma filtering surgery may be a promising technique for increasing bleb survival and reducing fibrosis [183].

#### 3.3.3. Sodium Hyaluronate

Healaflow is a cross-linked sodium hyaluronate that is absorbed slowly to prevent scarring, though it is not currently available in the US. Multiple human clinical studies have shown that Healaflow is a safe adjunct in both trabeculectomy and deep sclerectomy, and some studies have found that Healaflow enhances efficacy compared to these procedures without Healaflow [48,51,52,53,54,55]. However, in an RCT, Mudhol et al. showed that Healaflow and low-dose MMC are equally efficacious [52].

#### 3.3.4. Collagen Matrix

Ologen is a biodegradable three-dimensional collagen-glycosaminoglycan copolymer matrix implant that may also act as a physical spacer [56]. In an RCT, Ologen was shown to improve IOP control and blunt the hypertensive phase in Ahmed glaucoma valve surgery immediately after surgery but not at 6 or 12 months post-surgery [184]. Although early clinical studies comparing the rates of complications and efficacy of Ologen to MMC in trabeculectomy showed mixed results, recent trials have shown the comparable or superior efficacy of Ologen [56]. When comparing Ologen and perfluoropropane in trabeculectomy, the IOP values were similar in the first 12 months of follow-up, but the Ologen group showed a more significant reduction in IOP during the last 24 months of follow-up [50].

### 3.4. Modulating the Hypertensive Phase

The hypertensive phase, an increase in IOP that occurs usually in the first month postoperatively, is common following glaucoma filtration surgery. A higher preoperative IOP and younger age are risk factors for the hypertensive phase [185,186]. Pharmacological agents aimed to decrease the occurrence of the hypertensive phase include prostaglandin analogs and aqueous suppressant treatment [187]. However, recent findings suggest that the use of aqueous suppressants and prostaglandin analogs may affect wound healing. Aqueous suppressants have been shown to decrease the expression of α-SMA, reduce the transformation of fibroblasts to myofibroblasts, attenuate growth factors, and reduce mechanical stress from aqueous pressure on the bleb [187]. Prostaglandin analogs have been shown to induce collagen contraction, upregulate IL-2, and upregulate MMP-9. Jung et al. also showed that early treatment with aqueous suppressants decreased fibrosis in the bleb but early treatment with prostaglandin analogs did not decrease fibrosis. IOP at 4 weeks post-op was also lower in the aqueous suppressant group.

In current clinical practice, the most commonly used agents to prevent fibrosis and improve surgical outcomes are corticosteroids and antimetabolites. Most glaucoma specialists in the United States utilize MMC at the time of glaucoma filtration surgery (e.g., trabeculectomy), either via subconjunctival injection or soaked pledgets. The concentration of MMC and length of application of MMC varies by surgeon and patient characteristics. 5-FU as an antifibrotic is more commonly used intraoperatively in the United Kingdom. Postoperatively, topical corticosteroids are the mainstay of treatment to suppress ocular fibrosis. Typically, surgeons will prescribe topical prednisolone acetate 1% every 2 h while awake (8 times per day) or difluprednate 0.05% every 4 h while awake (4 times per day) for the first week and sometimes the first month after surgery before beginning to taper. Most surgeons do not utilize NSAIDs to mitigate fibrosis after glaucoma filtration surgery.

Overall, there is a great need for more advancements in our understanding of emerging and repurposed treatments that target fibrosis. Further investigation into solutions that will carry these molecular techniques from bench to bedside will allow for the ultimate goal of modulating fibrosis in glaucoma surgery, which will prevent vision loss among millions of people worldwide.

## 4. Novel Delivery Techniques and Biomaterials

Delivery methods of antifibrotic agents are important due to their rapid clearance from the subconjunctival space. It is important to prevent clearance while also ensuring that additional material–tissue interactions and localized foreign body responses are minimized. To prevent rapid clearance of antifibrotics, sustained-release implants, sponges, and injections have been used in certain therapies.

Hydrogels are a network of cross-linked polymer chains that are highly absorbent [188]. They are capable of increasing drug residence time and sustaining drug delivery to biomechanically modulate ocular wound healing. Many drugs with the potential for preventing ocular fibrosis have been tested in hydrogels for sustained release, including 5-FU, MMC, ranibizumab, and dexamethasone [115,188,189,190,191]. Hydrogels can provide specific mechanical signaling cues by acting as scaffolds for the infiltrating restorative cells, which makes them a promising approach for spatiotemporal control of ocular fibrosis.

Liposomal delivery systems have also been developed as a method of prolonging drug levels in the eye. Drug levels of 5-FU and mitoxantrone have been studied in rabbit models of glaucoma surgery, which have shown increased conjunctival drug concentrations and reduced ocular side effects [192,193].

Encapsulation of therapies in nanoparticles can enhance permeation across cell membranes and prevent enzymatic degradation [193]. Recent studies have shown that continuous nanoparticle drug delivery to targeted tissues and cells can be more effective than traditional drug delivery methods [194,195]. LDL receptors are overexpressed in activated Tenon’s fibroblasts. LDL-MMC nanoparticles specifically bind to upregulate LDL receptors on activated fibroblasts and are a novel therapy due to their selective targeting, small drug dose requirement, better bioavailability, and reduced cellular toxicity [196].

Alternative modes of drug delivery have been studied to increase the specificity of drug targets. Implanted biomaterials initiate nonspecific protein adsorption to the material surface, forming a matrix that is conducive to cellular adhesion. Biomaterial physical properties such as size, surface topography, and porosity influence fibroblast behavior and are critical to maintaining longevity and surgical efficacy [197].

There are many novel implants with biomaterials that are being studied in animal models that are promising for increasing the efficacy of glaucoma surgery. Electrospinning allows for the incorporation of polymers into nanofibers that can be manipulated into various combinations. Nano-structured glaucoma shunts are promising and have been tested in a rabbit model; they showed immediate IOP reduction and prevented post-op hypotony [198].

Dendrimers, hyperbranched macromolecules, can be engineered to a precise structure. They can be synthesized with anti-inflammatory properties. In vitro experiments with dendrimers developed by Shaunak et al. were found to reduce synthesis of pro-inflammatory chemokines and inhibit angiogenesis, in addition to increasing glaucoma filtration surgery success, in a rabbit model filtration surgery [199].

A custom-tailored expanded polytetrafluoroethylene (ePTFE) glaucoma drainage implant is another novel approach. The ePTFE glaucoma drainage implant is promising due to its high biocompatibility and inertness. It was found to reduce capsule thickness in blebs and increase their permeability compared with silicone implants in rabbits [199,200]. Collagen deposition is substantially lower with the ePTFE device than with the silicon implant control after one to three months. This device is currently being studied in human clinical trials, and the data are planned for release in 2023 [98].

Poly(styrene-block-isobutylene-block-styrene) (SIBS) is a stable, conforming, and biologically inert biomaterial that has been combined with the glaucoma drainage microtube as the PreserFlo^®^ MicroShunt [201]. SIBS results in virtually no foreign body reaction, which translates to insignificant inflammation and capsule formation. SIBS is still in clinical studies but shows promising efficacy and safety outcomes, although there is not yet enough evidence to recommend it over the current guidelines.

The Beacon Aqueous Microshunt shunts aqueous humor directly to the tear film [202]. Its channel is composed of polymerized, superhydrophilic polyethylene glycol, which prevents protein adsorption and bacterial attachments. This device is still in clinical trials.

In summary, various delivery techniques may significantly improve the glaucoma filtration surgery success rate. Many of these avenues of possibility in new drug delivery systems have not been fully characterized but have the potential for fine-tuning optimal drug delivery times, reducing cellular toxicity, and increasing the precision of drug delivery. Prolonging the presence of antifibrotic agents and utilizing novel biomaterials that attenuate the inflammatory response have significant implications in the treatment of glaucoma, allowing for greater IOP control and reducing post-op complications.

## 5. Conclusions

Several agents, devices, and biomaterials are potential alternatives to the current anti-fibrotic therapies, but they remain unavailable for clinical use. Many of the methods for the prevention of subconjunctival and sub-Tenon’s fibrosis have not been tested in large preclinical in vivo animal models and human RCTs. Retrospective case series and nonrandomized prospective studies provide insight into the potential for new glaucoma procedures and glaucoma filtration surgery adjuncts, but these early results should be interpreted with caution. Relevant evidence for the potential of new biomaterials and gene therapies is limited but is accumulating, as many are still in the early stages of testing. Likewise, several novel drug delivery approaches for antifibrotics are promising, but translation of these technologies to the bedside can be challenging. This review has described potential therapeutic targets for the prevention of ocular fibrosis after traditional incisional glaucoma filtering surgery, but RCTs are needed to improve our understanding of the long-term efficacy and safety of these novel therapies and to further define their role in clinical practice.

## Figures and Tables

**Figure 1 biomedicines-11-00657-f001:**
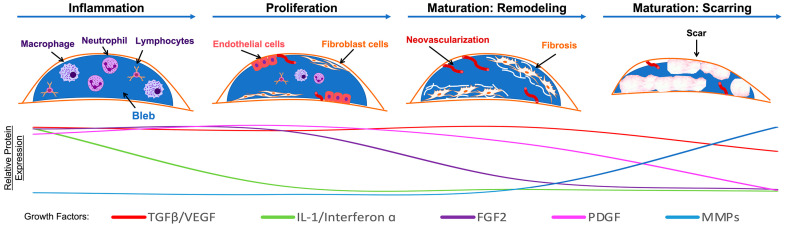
Stages of wound healing and key markers following glaucoma filtration surgery.

**Table 1 biomedicines-11-00657-t001:** Clinical, in vivo, and in vitro therapeutic agents and molecular targets for improving the success of glaucoma filtration surgery.

Clinical Studies						
Therapeutic Agent	Molecular Target	Route/Type of Delivery		Results	Potential Risks	Ref. No
Fluorouracil (5-FU)	Inhibits DNA synthesis	PO SC Inj.		Anti-proliferative Reduced filtration surgical failure	Hypotony maculopathy Bleb leak	[39,40,41]
Mitomycin-C (MMC)	Inhibits cell cycle, DNA replication, cell mitosis, and protein synthesis	SC Inj. Sponge ST Inj.		Reduced scar formation Increased bleb survival Rate Maintained reduced IOP	Bleb leakage Hypotony Choroidal effusion Endophthalmitis	[42,43,44,45,46]
Beta irradiation	Increases p53	Radiation		Minimized scar tissue formation Maintained reduced IOP	Cataract formation Keratopathy	[47,48]
Fibrinolytic therapy	Tissue plasminogen activator	Intracameral		Dissolved fibrin in blood clots	Hyphemia Profound hypotony Anterior chamber flattening	[49]
Perfluoropropane (C3F8)	Functional spacer	SC Inj.		Maintained reduced IOP		[50]
Healaflow	Cross-linked sodium hyaluronate	Surgical Implant		Maintained reduced IOP		[48,51,52,53,54,55]
Ologen	Biodegradable 3D COL-GAG copolymer matrix implant	Surgical Implant		Maintained reduced IOP		[56]
**In vivo Studies**						
**Therapeutic agent**	**Molecular Target**	**Route/type of Delivery**	**Ocular Fibrosis** **Model**	**Results**	**Potential** **Risks**	**Ref. No**
Fluorouracil (5-FU)	Inhibits DNA synthesis	Topical Eye Drops	Rabbit	Anti-proliferative Reduced filtration surgical failure	Toxic to actively replicating tissue	[57,58]
SMAD	Overexpress SMAD7 Reduces SMAD2, 3, 4	MicroRNAs (miRNA)	Rabbit	Reduced stromal fibrosis Modulated α-SMA expression		[26,59,60]
Anti-SPARC	SPARC	Nanoparticle-based-siRNA	Mouse Rabbit	Less collagenous ECM Smaller fibril collagen diameter Improved surgical survival		[61,62]
Pirfenidone	Inhibits TGF-β Reduces TNFɑ	Topical Eye Drops	Rats Rabbits	Inhibited extracellular matrix deposit Improved bleb survival		[63,64]
Anti-PDGF Subunit B agents	Binds PDGF	ARC126/127 E10030	Mouse	Anti-angiogenic Inhibited tumor growth		[65,66]
Anti-VEGF Agents	Inhibits VEGF	SC Inj. (Bevacizumab)	Rabbit Mouse	Reduced IOP Reduced angiogenesis Reduced collagen deposition	Scleral blebbing	[67]
MMC + Saratin	Inhibits cell cycle + prevents platelet aggregation	SC Inj. + Topical application	Rabbit	Increased bleb survival with a vascular bleb and tissue thinning		[68]
Lysyl Oxidase-Like 2	Collagen and elastin cross-linking	Inhibitory monoclonal antibody	Rabbit	Reduced pathologic angiogenesis, inflammation, and fibrosis		[69]
MMP	Degrades extracellular matrix	Topical Inhibitor SC IIlomastat	Rabbit	Improved bleb survival Reduced IOP Reduced scar tissue		[70,71]
MRTF	Fibroblast activation	Nanoparticle siRNA	Rabbit	Doubled bleb survival Decreased conjunctival scarring		[72]
SAHA	Histone modifier	SC Inj.	Rabbit	Improved bleb survival		[73]
p53 inhibitors	Tumor suppressor gene	AAV	Rabbit	Reduced IOP Prevented fibrosis proliferation		[74]
CDKN1B	Cell division pathway inhibitor	Topical Inhibitor	Rabbit	Reduced scar severity Improved surgical outcomes		[75]
Cyclosporine	Immunosuppressive	Topical Application	Rabbit	Enhanced effectiveness of glaucoma drainage		[76]
amniotic membrane	Anti-inflammatory, antifibrotic, antiangiogenic	Transplantation	Rabbit	Improved bleb survival	Delayed wound healing	[77]
Perfluoropropane (C3F8)	Functional spacer	SC Inj.	Rabbit	Maintained reduced IOP		[78]
**In vitro Studies**						
**Therapeutic agent**	**Molecular Target**	**Cell Type**		**Results**		**Ref. No**
5-FU	Antagonizes pyrimidine metabolism	Epithelial Fibroblast		Anti-proliferative Inhibited fibroblast-mediated collagen contraction		[79]
MMC	DNA crosslinking alkylating agent	Fibroblast		Inhibited fibroblast proliferation		[80,81,82]
Beta Irradiation	Increased p53	Fibroblast		Diminished fibroblast proliferation Targeted ECM production		[83,84]
Anti-TGF-β	Blocks TGFβ	Fibroblast		Mitigated fibroblast transdifferentiation		[85,86]
YAP/TAZ	Activates SMAD2/3 TGF-β1 and 2 mediators	Fibroblast		Reduced scaring markers		[87,88]
SPARC	Upregulates TGFβ2	Fibroblast		Modulated ECM production and organization		[89,90]
TGIF 1 and 2	Repress SMAD-dependent signaling	Fibroblast		Attenuated myofibroblast formation		[91,92,93,94]
Pirfenidone	Inhibits TGF-β Reduces TNFɑ	Epidural cell Fibroblast		Inhibited extracellular matrix deposit Decreased fibroblast proliferation, migration, and contraction		[95,96]
Nintedanib	Inhibits PDGF, VEGF, and FGF receptor	Fibroblast		Potent antifibrotic effect		[97,98]
Saratin	Collagen-Platelet interaction inhibitor	Platelet		Limited PDGF, TGFb, IGF, and EGF expression Prevented platelet aggregation		[99]
MMP	Degrades extracellular matrix	Fibroblast		Reduced collagen contraction, cell migration, and collagen production		[100]
MRTF	Fibroblast activation	Fibroblast		Blocked collagen matrix contraction		[101]
p53 inhibitors	Tumor suppressor gene	Fibroblast		Inhibited proliferation and DNA synthesis		[102,103]

## Data Availability

Not applicable.

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
