# Peer review of "Novel Therapies for the Prevention of Fibrosis in Glaucoma Filtration Surgery"

_biomedicines, 2023, doi:10.3390/biomedicines11030657_

Round 1

Reviewer 1 Report

The review article entitled “Novel Therapies for the Prevention of Fibrosis in Glaucoma 2 Filtration Surgery” by Christine et al, summarized the strategies and current state of promising future approaches for targeting fibrosis in glaucoma filtration surgery. The topic is original in the field. I have some comments:

There actually are other aspects that should also be included in this review. These include recent advances and innovations in ophthalmic wound healing research, including antibodies, RNAi, gene therapy, nanoparticles, liposomes, dendrimers, proteoglycans and small molecule inhibitors. This review article has 5 sections with a clear logical flow.

The graphical abstract only shows the well-known anatomy and pathology but does not show any potential therapies.

Figure 1 apart from showing stages of wound healing and key markers following glaucoma filtration surgery, can you add relevant information to link novel therapies for the Prevention of Fibrosis in Glaucoma 2 Filtration Surgery?

Reviewer 2 Report

Interesting review! Well described. Otter comments in the  attached file!

Good work!
